# Ultrasonic Evaluation of Diaphragm in Patients with Systemic Sclerosis

**DOI:** 10.3390/jpm13101441

**Published:** 2023-09-27

**Authors:** Anja Ljilja Posavec, Stela Hrkač, Josip Tečer, Renata Huzjan Korunić, Boris Karanović, Ivana Ježić, Ivan Škopljanac, Nevenka Piskač Živković, Joško Mitrović

**Affiliations:** 1Polyclinic for Respiratory Diseases, 10000 Zagreb, Croatia; 2School of Medicine, University of Zagreb, 10000 Zagreb, Croatia; 3Division of Clinical Immunology, Allergology and Rheumatology, Department of Internal Medicine, Dubrava University Hospital, 10000 Zagreb, Croatia; 4Clinical Department of Diagnostic and Interventional Radiology, University Hospital Dubrava, 10000 Zagreb, Croatia; 5Division of Clinical Immunology and Rheumatology, Department of Internal Medicine, University Hospital Centre Zagreb, 10000 Zagreb, Croatia; 6Department of Pulmonology, University Hospital Centre Split, 21000 Split, Croatia; ivan.skopljanac@gmail.com; 7Special Hospital Radiochirurgia Zagreb, 10000 Zagreb, Croatia; 8Faculty of Pharmacy and Biochemistry, University of Zagreb, 10000 Zagreb, Croatia

**Keywords:** diaphragm, systemic sclerosis, computed tomography, ultrasound, interstitial lung disease, lung function tests

## Abstract

The diaphragm is the most important muscle in respiration. Nevertheless, its function is rarely evaluated. Patients with systemic sclerosis (SSc) could be at risk of diaphragmatic dysfunction because of multiple factors. These patients often develop interstitial lung disease (SSc-ILD) and earlier studies have indicated that patients with different ILDs have decreased diaphragmatic mobility on ultrasound (US). This study aimed to evaluate diaphragmatic function in SSc patients using US with regard to the ILD, evaluated with the Warrick score on high-resolution computed tomography (HRCT), and to investigate associations between ultrasonic parameters and dyspnea, lung function, and other important clinical parameters. In this cross-sectional study, we analyzed diaphragm mobility, thickness, lung function, HRCT findings, Modified Medical Research Council (mMRC) dyspnea scale, modified Rodnan skin score (mRSS), autoantibodies, and esophageal diameters on HRCT in patients with SSc. Fifty patients were enrolled in the study. Patients with SSc-ILD had lower diaphragmatic mobility in deep breathing than patients without ILD. The results demonstrated negative correlations between diaphragmatic mobility and mMRC, mRSS, anti-Scl-70 antibodies, esophageal diameters on HRCT, and a positive correlation with lung function. Patients with SSc who experience dyspnea should be evaluated for diaphragmatic dysfunction for accurate symptom phenotyping and personalized pulmonary rehabilitation treatment.

## 1. Introduction

Systemic sclerosis (SSc) is an autoimmune disease of unknown etiology that can affect multiple organs and is characterized by the diffuse thickening of the skin [1]. Organ involvement is caused by interstitial and perivascular fibrosis, which can result in diffuse pulmonary disease, gastrointestinal malfunction, pulmonary hypertension, and renal involvement, among other things. Esophageal affection is highly prevalent in patients with SSc and presents with a variety of clinical symptoms, greatly affecting their quality of life [2]. In addition to clinical evaluation, specific antibodies, namely anticentromere and anti-topoisomerase I (anti-Scl-70), aid in the diagnosis of the disease [3]. One of the most common causes of mortality in patients with SSc is disease-associated interstitial lung disease (SSc-ILD) [4,5]. The most commonly reported symptoms of this disease are dyspnea and a non-productive cough [6]. Patients with SSc-ILD often have reduced lung function tests (forced vital capacity and lung diffusing capacity for carbon monoxide (DLCO)), which, combined with high-resolution computed tomography of the thorax (HRCT), are diagnostic methods for diagnosing and monitoring lung disease. HRCT of the thorax is the gold standard for diagnosing SSc-ILD [7]. One of the important scores used to evaluate SSc-ILD is the Warrick score [8,9,10,11]. A Warrick score ≥7 is predictive of abnormal lung function tests and indicates significant ILD [10]. The diaphragm is the most important muscle in respiratory physiology, responsible for more than 70% of minute ventilation [12]. When the diaphragm contracts, the tension in its fibers increases resulting in the descent of its dome [13].

The diaphragm separates the abdominal from the thoracic cavity and consists of the crural and costal part. The lower esophageal sphincter and the crural diaphragm are anatomically and functionally placed on top of each other, both preventing gastroesophageal reflux [14,15].

Diaphragm function can be reduced in some cardiac, pulmonary, neuromuscular, infectious, and autoimmune diseases [16,17,18,19,20]. Its function is not routinely evaluated in clinical practice, and data are scarce regarding the prevalence of diaphragmatic dysfunction, which can be one of the causes of dyspnea and cough, but often remains unrecognized. Fluoroscopy, electromyography, and esophagomanometry are methods used to evaluate diaphragmatic function. However, they are rarely applied in everyday practice. These diagnostic methods are often unsuitable because they expose patients to radiation or because of their invasiveness. In patients with a paralyzed diaphragm, a chest X-ray can reveal an elevated hemidiaphragm [21]. However, the elevation of both hemidiaphragms has low specificity for diagnosing diaphragmatic dysfunction. Although a chest X-ray is useful for detecting unilateral diaphragmatic paralysis, its specificity is unsatisfactorily low [22].

We hypothesized that patients with SSc-ILD could be at risk for weaker diaphragm function. Increased lung tissue resistance, among other factors, could cause lower diaphragm mobility. Other possible causes for diaphragm dysfunction among SSc patients that could affect diaphragm contractility could be disease severity, inflammation, glucocorticoids use, sedentary lifestyle, malnutrition, etc.

Ultrasound (US) is increasingly used in the evaluation and screening of patients with ILDs of different origins [23]. Pneumothorax, pleural effusion, the consolidation of lung parenchyma, interstitial syndrome, and other conditions can also be diagnosed using US [24].

Additional information obtained during US examination with screening for diaphragmatic dysfunction would be useful in the evaluation of dyspnea in SSc patients. Diaphragmatic dysfunction can be difficult to diagnose in the absence of complete paralysis and in the absence of clinical suspicion. An US of the diaphragm is a non-invasive, available, and portable method that can be safely used in daily practice to evaluate diaphragmatic function. The diaphragmatic thickening fraction (DTF) indicates whether appropriate diaphragm thickening occurs during inspiration [25]. So far, there is no expert consensus regarding the reference values for diaphragmatic mobility and DTF in a healthy population because of the differences in the techniques used [26]. A lack of increment in diaphragmatic thickening during inspiration is a more sensitive and specific factor for diaphragmatic paralysis than the measurement of diaphragmatic thickness itself [27]. The diaphragm, the most important respiratory muscle, has not yet been investigated in patients with SSc with US assessment. The primary aim of this study was to evaluate diaphragmatic function during the respiratory cycle in patients with SSc and to evaluate whether patients with SSc-ILD have decreased parameters of diaphragm mobility and contractility. The secondary goal was to investigate associations between diaphragmatic parameters and other relevant clinical markers in patients with SSc, such as lung function tests, Modified Medical Research Council dyspnea scale (mMRC), modified Rodnan skin score (mRSS), autoantibodies, and esophageal diameters on HRCT in patients with SSc.

The measurement of skin thickness by mRSS is used as a marker for disease activity and severity in patients with diffuse SSc [28]. Esophageal diameters on HRCT have been evaluated in earlier studies, and results have pointed to the relationship between SSc-ILD and esophageal dilatation on HRCT [11,29].

## 2. Materials and Methods

### 2.1. Study Design

This was a multicenter observational cross-sectional study. The study is reported in line with the STROBE reporting guideline for cohort studies Appendix A.

### 2.2. Patient Characteristics

The subjects in this study were adult patients diagnosed with SSc who were regularly followed up by a rheumatologist at the Internal Clinic of the University Hospital Dubrava, the Department of Clinical Immunology, Allergology, and Rheumatology, and at the Department of Clinical Immunology and Rheumatology at the University Hospital Centre, Zagreb. Patients were consecutively included in the study between July 2021 and January 2023. The study received ethical approval from the Research Ethics Boards. The inclusion criteria were age 18–80 years, fulfillment of the 2013 ACR/EULAR classification criteria for SSc [3], and written consent for participation in the study. Exclusion criteria were acute heart failure, pleural effusion, respiratory infection or lung disease that is not specific to patients with SSc, other autoimmune or neurological diseases that could contribute to impaired diaphragmatic mobility, and a body mass index (BMI) <18.5 and >40. Clinical data, such as disease duration, SSc-related antibodies, and comorbidities, were also recorded and analyzed. The mRSS score was measured by estimating skin hardening (from 0 to 3) in 17 regions of the body [30,31]. Additionally, gastroesophageal symptoms, heartburn, gastric acid/content regurgitation, dysphagia, and/or epigastric pain, were noted during the examination. Figure 1 represents a study outline describing the parameters evaluated in the patient groups.

### 2.3. Evaluation of Lung Disease

Lung disease was evaluated using pulmonary function tests and the standardized scoring of HRCT findings and dyspnea (Figure 1). Spirometry and DLCO were performed according to ATS/ERS standards (EasyOne Pro, 651025, ndd Medizintechnik AG, Zürich, Switzerland) [32,33]. The criteria for SSc-ILD were determined according to the HRCT findings interpreted by the same experienced thoracal radiologist using the Warrick score, Appendix A [34]. The radiologist was blinded to the results of the lung function tests. According to the Warrick score, five types of lesions were evaluated for severity score. The number of affected lung segments was evaluated for the extension score, while the total sum of both equaled the global Warrick score [8,10].

Dyspnea was evaluated using the mMRC dyspnea scale to measure breathlessness in everyday activities on a scale from 0 to 4: 0—no breathlessness except with strenuous exercise; 1—shortness of breath while hurrying on the level or walking up a slight hill; 2—patient walks slower than people of the same age on the level because of breathlessness/has to stop when walking at their own pace on the level; 3—patient stops for breath after walking 100 m; and 4—patient is too breathless to leave the house/breathless while dressing or undressing [35,36,37].

A Warrick score ≥ 7 was used as a cut-off value for significant ILD [10].

### 2.4. Evaluation of the Diaphragm

US was used to assess the diaphragm function. In the supine position of the patient, diaphragmatic thickness was measured in B mode with a high-frequency linear probe of 10 MHz (Philips Affiniti 70 ultrasound machine, Philips Ultrasound Inc., Bothell, WA, USA) in the state of functional residual capacity (FRC) and total lung capacity (TLC). A linear probe was placed in the anterior axillary line between the 7th and 9th intercostal spaces in the zone of apposition in the longitudinal plane.

During contraction, this area normally shortens and thickens.

The diaphragm was visualized as a three-layer structure consisting of the diaphragmatic pleura and peritoneal membrane with hypoechoic diaphragmatic muscle in-between [23]. The distance in echogenic lines between the diaphragmatic pleura and peritoneal membrane was measured in frozen images [38].

The diaphragmatic thickening fraction was calculated using the following formula: (thickness at the end of the maximum inspiration–thickness at the end of the expiration)/thickness at the end of the expiration × 100 [23,39,40]. Diaphragm thickness at a FRC < 0.15 cm was considered decreased [40].

Using a convex ultrasound probe 2–6 MHz in B mode (in abdominal preset), with an anterior subcostal approach, between the midclavicular and anterior axillary line, the right diaphragm was visualized. The probe was directed medially, cephalically, and dorsally so that the US beam reached the right dome perpendicularly. Subsequently, when a good quality of the image was obtained, the amplitude of craniocaudal diaphragmatic mobility during normal/tidal and deep breathing was measured in the M mode [41,42]. The highest values of three measurements were selected. The mobility of the diaphragm was measured only on the right side because the left hemidiaphragm often cannot be adequately visualized, which is a common limitation of the ultrasonic method [43,44].

The investigator who was performing all examinations was blinded to lung function results and HRCT findings.

### 2.5. Evaluation of the Esophagus

HRCT scans were used to evaluate the esophageal dimensions (diameter). The greatest distance between the inner border of the esophageal mucosa was measured at three points on axial HRCT scans: above the aortic arch (location 1), between the right inferior pulmonary vein and the aortic arch (location 2), and between the diaphragmatic hiatus and the right inferior pulmonary vein (location 3) [11,29].

### 2.6. Sample Size Calculation

Based on previous research [45], and to observe differences with a significance level of 0.05 and a test power of 80%, a minimum of 48 patients were required (G*Power ver. 3.1. 9.2).

### 2.7. Statistical Methods

Categorical data were presented by absolute and relative frequencies. The normality of the distribution of continuous variables was tested using the Shapiro–Wilk test. Continuous data were described by the median and the limits of the interquartile range (IQR). The Mann–Whitney U test was used to compare the median between two groups, while the Fisher’s exact test was used to analyze the differences between proportions. The Spearman’s rho test was used to determine the association between non-normally distributed variables. Regression analysis (bivariate and multivariate) was used to determine the influence of individual predictors on SSc-ILD. All *p* values were two-sided. The significance level was set at an alpha value of 0.05. The statistical analysis was performed using MedCalc^®^ Statistical Software version 22.006 (MedCalc Software Ltd., Ostend, Belgium; https://www.medcalc.org; 16 September 2023) and IBM SPSS Statistics for Windows, ver. 29 (IBM Corp., Armonk, NY, USA).

## 3. Results

### 3.1. Patients and Characteristics

Fifty patients diagnosed with SSc were consecutively included in the study after meeting the inclusion and exclusion criteria. Initially, fifty-three patients were included, but three patients were excluded from the study. One patient due to acute heart failure, a second patient due to the disease overlapping with polymyositis, and a third patient due to cognitive deterioration, meaning the patient could not follow the instructions for the ultrasonic procedure and lung function testing.

Significant SSc-ILD with a global Warrick score of ≥7 was present in 30 (60%) patients.

Table 1 shows the demographic and clinical characteristics of patients. Patients with SSc-ILD more frequently had pulmonary hypertension, gastroesophageal symptoms, and wider esophageal diameters on HRCT.

There were no significant differences in the other evaluated comorbidities, such as arterial hypertension, diabetes mellitus, hypothyroidism, and malignant disease, between the two groups. The median global Warrick score in patients with SSc-ILD was 12 (IQR 8–15). The median severity Warrick score was 6 (IQR 4–6) and the extent score was 6 (IQR 4–9).

### 3.2. Results of Ultrasonic Diaphragmatic Assessment

Table 2 demonstrates the values of ultrasonic assessment of diaphragm mobility, thickness, and DTF among all study participants and separately for groups of patients with and without SSc-ILD. Patients with SSc-ILD had lower diaphragmatic mobility in deep breathing (*p* = 0.004). There was no statistically significant difference in normal/tidal breathing between the two groups. There were no significant differences in diaphragm thickness or diaphragmatic thickening fraction between the two groups.

Three (6%) patients with SSc had reduced values of diaphragm thickness at FRC (0.14 cm) with regard to cut-off values < 0.15 cm [40].

### 3.3. Correlations of Diaphragmatic Mobility with other Evaluated Clinical Parameters

There was a weak negative correlation between diaphragm mobility in deep breathing and the mMRC scale (r = −0.286, *p* = 0.04), presence of anti-Scl-70 antibodies (r = −0.305, *p* = 0.03), esophageal diameters on HRCT locations 1 and 2 (r = −0.368, *p* = 0.01 and r = −0.366, *p* = 0.01, respectively) (Appendix A), and mRSS score (r = −0.299, *p* = 0.04) (Appendix A).

Diaphragm mobility during deep breathing correlated with FVC% and VA% (r = 0.313, *p* = 0.03, and r = 0.501, *p* < 0.001, respectively) (Appendix A, Appendix A), and with the presence of anticentromere antibodies (r = 0.339, *p* = 0.02).

### 3.4. Factors Predicting SSc-ILD

In bivariate regression analysis, SSc-ILD was associated with gastroesophageal symptoms (odds ratio [OR] = 7.54, 95% CI = 1.37–41.4), and greater esophageal diameters at location 1 (OR = 1.76, 95% CI = 1.19–2.6), location 2 (OR = 1.27, 95% CI = 1.09–1.47), and location 3 (OR = 1.19, 95% CI = 1.06–1.35) on HRCT. Multivariate regression analysis demonstrated that an independent factor predicting SSc-ILD was esophageal diameter on HRCT at location 1 (OR =1.76, 95% CI= 1.19–2.60) (Appendix A). This model accurately predicted SSc-ILD in 71% of the patients.

## 4. Discussion

Diaphragmatic dysfunction is frequently underrecognized [46]. Two sonographic techniques are used for the diaphragmatic assessment: the muscle thickening in the zone of apposition and excursion of the diaphragm during respiration.

The research results demonstrate that patients with SSc-ILD have lower diaphragmatic mobility in deep respiration than patients without SSc-ILD.

Our results correlate with earlier studies on diaphragm mobility in patients with ILD, but with different origins of lung diseases (such as fibrotic hypersensitivity pneumonitis, idiopathic pulmonary fibrosis, etc.) in which patients with ILD also had lower DTF than healthy controls [45,47]. In our study, no statistically significant difference was observed between the diaphragmatic thickness and DTF between the groups of patients with and without SSc-ILD.

Diaphragm mobility was in positive correlation with lung function test results, FVC%, VA%, and in negative correlation with the mMRC score. It is known that diaphragmatic dysfunction can contribute to the worsening of dyspnea, especially in patients with known cardiorespiratory comorbidities in SSc. It is possible that if selected patients are referred for pulmonary rehabilitation [48,49,50], the integral part of which is inspiratory muscle training, with concomitant treatment of the underlying disease, respiratory symptoms would decrease but further prospective studies are needed.

Patients with SSc in our study had median DTF values of 0.48 (IQR 0.33–0.67) or 48%, while the DTF values of healthy controls in the study by Santana were much higher (1.31 ± 55) [45]. A study by Spiesshoefer et al. confirmed that men have a greater craniocaudal excursion amplitude than women, greater diaphragm thickness, and no statistical significance between sexes for DTF. The lower limit of normal DTF in healthy subjects was 121%. Furthermore, lower limits of normal for craniocaudal excursion at TLC were 7.9 cm for men and 6.4 cm for women [26], while the patient groups in our study had much lower values of craniocaudal excursion with median values of 5.05 cm (IQR 2.43–6.71) in patients with SSc-ILD and 5.85 cm (IQR 4.5–7.03) in patients without ILD, which highlights significantly lower values.

Any of the patients with SSc could be at risk for reduced diaphragmatic function due to a sedentary lifestyle, therapy, hypoxic stress, inflammation, disease severity, malnutrition, etc. [51]. Malnutrition is a frequent complication of SSc. In the study by Rivet et al. on one hundred and twenty patients with SSc more than half of the patients were malnourished according to the 2020 French recommendations and 25% had at least one criterion of severe malnutrition [52]. The causes for malnutrition in SSc are multiple, such as gastrointestinal tract affection, inflammation and disease burden [53]. Sarcopenia, which is highly prevalent among patients with SSc [54], is associated with lower muscle strength and mass [55]. According to the World Health Organization (WHO) criteria, a body mass index (BMI) < 18.5 is a cut-off value for underweight. According to Kantarci et al., BMI values < 18.5 and greater than 40 were associated with statistically lower diaphragm mobility in healthy subjects [39]. Accordingly, these characteristics were the exclusion criteria in our study. Patients in our study had a median BMI of 25 without statistically significant differences between the two groups.

Potential further research on SSc patients could involve measuring the effects of sarcopenia and malnutrition on diaphragm muscle function.

Earlier studies have identified several disorders as possible causes of diaphragmatic dysfunction, such as neuromuscular diseases [56], stroke with hemiplegia [38,57], prolonged artificial ventilation [25,43], damage to the phrenic nerve, disuse atrophy, drugs, infectious or inflammatory conditions, chronic obstructive pulmonary disease (COPD) and hyperinflation [58], systemic lupus erythematosus [59,60], mixed connective tissue disease [61], but, so far, there is little evidence regarding diaphragm function in SSc patients, and further prospective and randomized studies should be encouraged. In our study, there was a wide age range among participants, considering that SSc is a rare orphan disease. There are a variety of possible causes for decreased diaphragm function in older people [51], but the results of van Doorn et al. suggest that diaphragm thickness is constant across a broad age range in healthy subjects [62], and, according to a study by Boon et al. on 150 healthy subjects, diaphragm thickening is minimally affected by age, sex, habitus, and smoking history [40].

SSc-ILD, as a consequence of the esophageal involvement of the disease, has been reported in many studies [9,63]. In our study, patients with SSc-ILD had significantly wider esophageal diameters and more gastroesophageal symptoms. Multivariate regression analysis demonstrated that an independent factor predicting SSc-ILD was esophageal diameter on HRCT at location 1, and this model accurately predicted lung disease in 71% of the patients.

The research results also demonstrated a negative correlation between diaphragm mobility in deep breathing and esophageal diameters on HRCT. It should also be noted that these relationships do not imply causality.

It is possible that patients with more severe disease have worse SSc-ILD and decreased respiratory muscle function. However, patients with ILD have increased elastic recoil of the lungs [64], which can cause decreased diaphragm mobility. An overlap of SSc with other connective tissue diseases such as dermatomyositis, polymyositis, Sjogren’s syndrome, systemic lupus erythematosus, and rheumatoid arthritis has an impact on the clinical picture and treatment [65,66]. This study included only patients affected with SSc, while further larger prospective studies on diaphragm function that would include overlapping connective tissue diseases should be encouraged.

As pulmonary function tests are not sensitive enough, and lifetime exposure to many HRCT scans can contribute to the appearance of malignant diseases, lung US has been the focus of evaluating SSc-ILD [67,68]. It can be used in diagnosing and monitoring lung disease. We would like to recommend US assessment of the diaphragm as an additive screening method to lung US for patients with unexplained or progressive dyspnea, and/or decreased lung volumes. Diaphragm evaluation can also be performed to select patients who may benefit from respiratory rehabilitation and for future follow-up, although further research is required to evaluate whether this will have a positive impact on the clinical endpoints. The results of this study provide insights into respiratory muscle function in patients with a serious and very rare autoimmune disease. We believe that a multidisciplinary and personalized approach is needed in this complex group of patients. This is the first research in the literature in which diaphragmatic function was evaluated in SSc patients with ultrasound.

### 4.1. Strengths

The strength of this research is that it is a multicentric study performed in two tertiary centers specialized in the treatment of SSc. Except for the evaluation of the diaphragm, an extensive evaluation of other clinical parameters was performed. SSc-ILD was evaluated using the Warrick score, which gives us relevant information on the extension and type of lung lesions.

The diaphragm was examined in the supine position of the patients which has less variability and greater reproducibility compared to other positions, and is the preferred positioning for diaphragmatic ultrasound [41]. The ultrasonic evaluation as well as interpretation of HRCT scans were performed by one investigator, and that way we eliminated inter-observer variability. Patients with only one type of ILD were included, in comparison to other studies on diaphragmatic US which evaluated ILDs of different origins.

We have proposed future research directions on diaphragmatic function evaluation in SSc patients.

### 4.2. Limitations

The limitations of our study include the relatively small number of patients, dependence on patient cooperation and effort during US assessment, lack of previous studies in the research area regarding diaphragmatic function in SSc patients, the cross-sectional study design, lack of healthy controls, and the wide age range for inclusion criteria. The mobility of the diaphragm was assessed only on the right hemidiaphragm because the left hemidiaphragm often cannot be adequately visualized, which is a common limitation of the ultrasonic method [43,44].

## 5. Conclusions

The research results demonstrate that patients with SSc-ILD have lower diaphragmatic mobility in deep breathing than patients without ILD. The results also demonstrated negative correlations between diaphragmatic mobility, mMRC scale score, and other relevant parameters such as mRSS, and esophageal diameters which altogether indicate a more severe clinical picture. Diaphragmatic mobility in deep respiration had a positive correlation with lung function. We propose US assessment of the diaphragm as a simple and noninvasive additive screening method for patients with SSc and dyspnea. Because diaphragm dysfunction can often be unrecognized, diaphragmatic US is a useful method for evaluating diaphragm function by assessing its mobility and contractility in order to objectify the possible causes of dyspnea. Further prospective randomized studies with larger sample sizes are needed to understand the complexity of SSc patients.

## Figures and Tables

**Figure 1 jpm-13-01441-f001:**
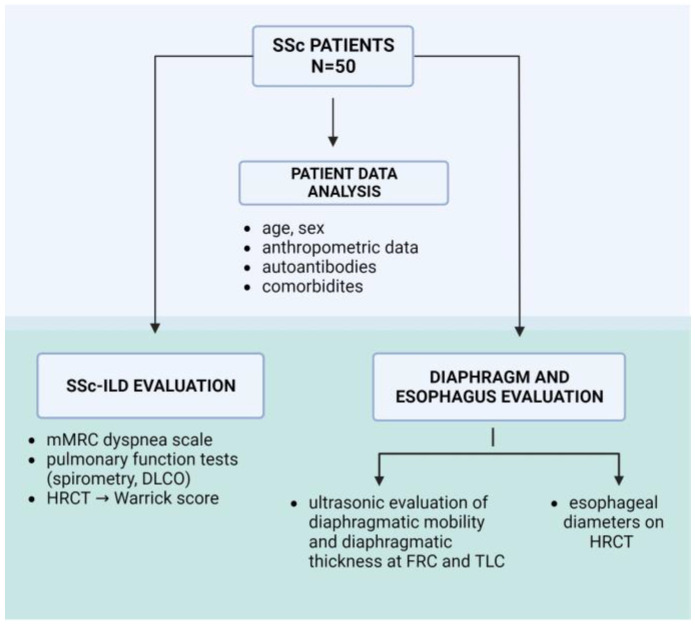
A study outline describing the different parameters evaluated in the patient groups, as well as the different diagnostic modalities used in the study. SSc-ILD—systemic sclerosis-associated interstitial lung disease, mMRC—Medical Research Council dyspnea scale, DLCO—lung diffusing capacity for carbon monoxide, HRCT—high-resolution computed tomography, FRC—functional residual capacity, TLC—total lung capacity.

**Table 1 jpm-13-01441-t001:** Characteristics of the patients with systemic sclerosis.

Characteristics of Patients with Systemic Sclerosis	All Patients (N 50)	Patients with Interstitial Lung Disease (N 30)	Patients without Interstitial Lung Disease (N 20)	*p* Value
Age	63 (54–70)	64 (54–72)	60 (46–65)	0.17 *
Gender: Male	5 (10)	4 (13)	1 (5)	0.64 ^†^
Female	45 (90)	26 (87)	19 (95)	
Duration of disease, years	9 (5–17)	13 (7–22)	6 (5–10)	0.01 *
BMI	25.5 (23.2–29.1)	24.3 (23.1–26.4)	28.27 (23.2–31.0)	0.10 *
Weight (kg)	68.5 (60–82.5)	65.5 (56.8–72.3)	78 (65–94.25)	0.05 *
Height (cm)	165 (158–170)	162 (158–170.3)	166 (165–170)	0.09 *
Anti-Scl-70 antibodies	18 (36)	14 (47)	4 (20)	0.07 ^†^
Anticentromere antibodies	16 (32)	6 (20)	10 (50)	0.03 ^†^
mRSS	6.5 (2–13)	9.5 (1.75–13)	4 (2–13)	0.38 *
Gastroesophageal symptoms	41 (82)	28 (93)	13 (65)	0.02 *
Pulmonary hypertension	10 (20)	9 (30)	1 (5)	0.04 ^†^
FEV1/FVC	0.79 (0.8–0.8)	0.79 (0.8–0.8)	0.76 (0.7–0.8)	0.12 *
FEV1 (%)	91 (81.8–101.8)	87 (77.3–99.5)	97.5 (87–104.8)	0.03 *
FVC (%)	98.5 (84–109.3)	91 (78–105.8)	102 (96.3–109.8)	0.03 *
MMEF (%)	70 (48–92)	64 (45–83)	84 (68.3–105.8)	0.02 *
DLCO (%)	69.5 (58.5–85.3)	64.5 (47.5–70.5)	81.5 (73–98.8)	<0.001 *
KCO (%)	80.5 (67.3–90.8)	76 (63–82)	87.5 (72.5–97.3)	0.01 *
VA (%)	90 (79–101)	84 (72.5–95)	97.5 (90–105)	0.002 *
Ground-glass opacities	28 (56)	26 (87)	2 (10)	<0.001 ^†^
Irregular pleura	25 (50)	22 (73)	3 (15)	<0.001 ^†^
Septal/subpleural lines	37 (74)	29 (97)	8 (40)	<0.001 ^†^
Honeycombing	5 (10)	5 (17)	0	0.08 ^†^
Subpleural cysts	7 (14)	7 (23)	0	0.03 ^†^
Esophageal diameter at location 1 (mm)	8 (6–14)	12 (7.5–16)	7 (6–8)	<0.001 *
Esophageal diameter at location 2 (mm)	12 (7.5–18.5)	17 (9.5–22.5)	8 (7–10.75)	<0.001 *
Esophageal diameter at location 3 (mm)	16 (10.5–21)	20 (13–25.5)	11 (9–16)	0.001 *

* Mann–Whitney U test; ^†^ Fisher’s Exact Test; BMI—body mass index; mRSS—modified Rodnan skin score; FEV1—forced expiratory volume in the first second; FVC—forced vital capacity; MMEF—mid-maximum expiratory flow; DLCO—diffusing capacity for carbon monoxide; KCO—carbon monoxide transfer coefficient; VA—alveolar volume.

**Table 2 jpm-13-01441-t002:** Ultrasonic assessment of the diaphragm mobility, thickness, and thickening fraction among patients with systemic sclerosis with respect to disease-associated interstitial lung disease.

	All Patients (N 50)	Patients with Interstitial Lung Disease (N 30)	Patients without Interstitial Lung Disease (N 20)	*p* Value
Diaphragm mobility in deep breathing (cm)				
	5.35 (3.75–6.88)	5.05 (2.43–6.71)	5.85 (4.5–7.03)	0.004
Diaphragm mobility in quiet breathing (cm)	1.59 (1.18–2.13)	1.8 (1.50–2.50)	1.45 (1–1.92)	0.21
Diaphragm thickness at FRC (mm)				
Right side	0.20 (0.20–0.30)	0.24 (0.20–0.30)	0.20 (0.17–0.30)	0.20
Left side	0.24 (0.20–0.30)	0.30 (0.20–0.30)	0.21 (0.18–0.30)	0.20
Diaphragm thickness at TLC (mm)				
Right side	0.40 (0.30–0.40)	0.40 (0.30–0.40)	0.36 (0.29–0.42)	0.80
Left side	0.36 (0.30–0.40)	0.38 (0.30–0.43)	0.34 (0.28–0.40)	0.51
Thickening fraction				
Right side	0.48 (0.33–0.67)	0.50 (0.33–0.67)	0.48 (0.33–0.84)	0.32
Left side	0.44 (0.33–0.65)	0.40 (0.33–0.50)	0.43 (0.33–0.80)	0.30

FRC—functional residual capacity; TLC—total lung capacity.

## Data Availability

The datasets analyzed during the study are available from the corresponding authors on reasonable request.

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
