# Peer review of "Ultrasonic Evaluation of Diaphragm in Patients with Systemic Sclerosis"

_jpm, 2023, doi:10.3390/jpm13101441_

Round 1

Reviewer 1 Report

In the manuscript entitled "Ultrasonic evaluation of diaphragm in patients with systemic sclerosis " the authors have evaluated diaphragmatic function during the respiratory cycle in 50 patients with systemic sclerosis (SSc) using ultrasonography. Additionally, they investigated associations between diaphragmatic parameters and other relevant clinical markers, such as lung function tests, Modified Medical Research Council dyspnoea scale (mMRC), modified Rodnan skin score (mRSS), autoantibodies, and oesophageal diameters on HRCT in patients with SSc. The patients with SSc were divided into two groups depending on the high resolution computed tomography (HRCT) findings and an assessment of severity and extent of lung disease using the Warrick score. Warrick score ≥7 was used as a cut-off value for significant interstitial lung disease (ILD).

The authors found that patients with SSc-ILD had lower diaphragmatic mobility in deep breathing than patients without ILD. Additionally, they showed that there was a weak negative correlation between diaphragm mobility in deep breathing and mMRC scale (r=-0,286, p=0,04), presence of anti-Scl-70 antibodies (r=-0,305, p=0,03), oesophageal diameters on HRCT locations 1 and 2 (r=-0,368, p=0,01 and r=-0,366, p=0,01, respectively), and mRSS score (r=-0,299, p=0,04). Moreover, diaphragm mobility during deep breathing correlated with FVC% and VA% (r=0.313, p=0.03, and r=0.501, p<0.001, respectively) and with the presence of anticentromere antibodies (r=0,339, p=0,02).

The manuscript is quite well-written and contains important information regarding the usefulness of the diaphragm ultrasonography in assessing the causes of dyspnea in patients with SSc as a simple non-invasive additive screening method.

However, there are some concerns regarding this paper as follows:

Major points:

1. In Results the authors showed the correlations of diaphragmatic mobility with some evaluated clinical parameters. I suggest to add the graphs showing the correlations, for example, as  a supplementary material.

 2. In the Discussion the authors stated that: ” Diaphragm mobility was in negative correlation with lung function tests results FVC%, VA% and with mMRC score.”  This statement is not consistent with the results presented in subsection 3.3 where the positive correlation was shown between diaphragm mobility and FVC% and VA%. The same inconsistencies apply to the statements in the Abstract and in the subsection Conclusions. I will ask for clarification/correction.

3. In the Conclusions subchapter the sentence : “We propose US assessment of the diaphragm as a simple non-invasive additive screening method while performing lung US for patients with SSc and dyspnoea to spare patients of unnecessary HRCT radiation”  is an over-reaching conclusion. In my opinion, such a conclusion cannot be formulated. Ultrasound examination of the diaphragm can be an additional, complementary examination to HRCT, but it cannot replace this examination in the search for the cause of dyspnoea.

4. Please reconsider the subchapter Conclusions and modify them as they do not quite correspond to the Results subchapter.

Reviewer 2 Report

The authors showed the results about Ultrasonic evaluation of diaphragm in patients with systemic sclerosis. The article is original, weel written and it contributes to the advancement of knowledge in this field.

My recommendations are:

-study design should be specified. Th enumber of excluded patients and reasons for exclusion should be moved tro the results section.

- Beyond BMI, malnutrition have been studied? I suggest to add in the discussin more information about reduced weight loss, malnutrition, sarcopenia and reduced muscle mass in systemic sclerosis patients.

-How is calculated the sample size?

- strenghts and weakness should be addedd

The authors showed the results about Ultrasonic evaluation of diaphragm in patients with systemic sclerosis. The article is original, weel written and it contributes to the advancement of knowledge in this field.

My recommendations are:

-study design should be specified. Th enumber of excluded patients and reasons for exclusion should be moved tro the results section.

- Beyond BMI, malnutrition have been studied? I suggest to add in the discussin more information about reduced weight loss, malnutrition, sarcopenia and reduced muscle mass in systemic sclerosis patients.

-How is calculated the sample size?

- strenghts and weakness should be addedd

Round 2

Reviewer 1 Report

Dear Authors,

Thank you for making changes to the manuscript in line with the reviewers' suggestions. I have no further comments. I accept the paper for publication.